# A Flavonoid-Rich Extract of *Sambucus nigra* L. Reduced Lipid Peroxidation in a Rat Experimental Model of Gentamicin Nephrotoxicity

**DOI:** 10.3390/ma15030772

**Published:** 2022-01-20

**Authors:** Rodica Ana Ungur, Ileana Monica Borda, Răzvan Andrei Codea, Viorela Mihaela Ciortea, Bogdana Adriana Năsui, Sevastița Muste, Orsolya Sarpataky, Miuța Filip, Laszlo Irsay, Elena Cristina Crăciun, Simona Căinap, Delia Bunea Jivănescu, Anca Lucia Pop, Victoria Emilia Singurean, Maria Crișan, Oana Bianca Groza, Georgiana Smaranda Martiș (Petruț)

**Affiliations:** 1Department of Medical Specialties, Faculty of Medicine, “Iuliu Hațieganu” University of Medicine and Pharmacy, 8 Victor Babeș Street, 400012 Cluj-Napoca, Romania; ungurmed@yahoo.com (R.A.U.); viorela.ciortea@yahoo.com (V.M.C.); laszlo.irsay@gmail.com (L.I.); 2Faculty of Veterinary Medicine, University of Agricultural Science and Veterinary Medicine, 3-5 Mănăștur Street, 400372 Cluj-Napoca, Romania; orsolya.sarpataky@usamvcluj.ro; 3Department of Community Health, Faculty of Medicine, “Iuliu Hatieganu” University of Medicine and Pharmacy, 8 Victor Babeș Street, 400012 Cluj-Napoca, Romania; nasuibogdana@yahoo.ro; 4Food Engineering Department, University of Agricultural Sciences and Veterinary Medicine, 64 Calea Floresti, 400509 Cluj-Napoca, Romania; sevastita.muste@usamvcluj.ro (S.M.); georgiana.petrut@usamvcluj.ro (G.S.M.); 5Raluca Ripan Institute for Research in Chemistry, Babeş-Bolyai University, 30 Fântânele Street, 400294 Cluj-Napoca, Romania; miuta.filip@ubbcluj.ro; 6Department of Pharmaceutical Biochemistry and Clinical Laboratory, Faculty of Pharmacy, “Iuliu Hatieganu” University of Medicine and Pharmacy, 6 Pasteur Street, 400349 Cluj-Napoca, Romania; cristina.craciun@umfcluj.ro; 7Department of Mother and Child, Faculty of Medicine, “Iuliu Hatieganu” University of Medicine and Pharmacy, 8 Victor Babeș Street, 400012 Cluj-Napoca, Romania; cainap.simona@gmail.com; 8Department of Internal Medicine, Faculty of Medicine, “Iuliu Hatieganu” University of Medicine and Pharmacy, 8 Victor Babeș Street, 400012 Cluj-Napoca, Romania; delia20002002@yahoo.com; 9Department of Clinical Laboratory, Food Safety, Nutrition, “Carol Davila” University of Medicine and Pharmacy, 6 Traian Vuia Street, 020945 Bucharest, Romania; anca.pop@umfcd.ro; 10Department of Morphological Sciences, Faculty of Medicine, “Iuliu Hatieganu” University of Medicine and Pharmacy, 8 Victor Babeș Street, 400012 Cluj-Napoca, Romania; tuns.victoria@umfcluj.ro (V.E.S.); mcrisan7@yahoo.com (M.C.); oana.groza@polarismedical.ro (O.B.G.)

**Keywords:** antioxidants, gentamicin, nephrotoxicity, elderflower, flavonoids, oxidative stress

## Abstract

The use of gentamicin (GM) is limited due to its nephrotoxicity mediated by oxidative stress. This study aimed to evaluate the capacity of a flavonoid-rich extract of *Sambucus nigra* L. elderflower (SN) to inhibit lipoperoxidation in GM-induced nephrotoxicity. The HPLC analysis of the SN extract recorded high contents of rutin (463.2 ± 0.0 mg mL^−1^), epicatechin (9.0 ± 1.1 µg mL^−1^), and ferulic (1.5 ± 0.3 µg mL^−1^) and caffeic acid (3.6 ± 0.1 µg mL^−1^). Thirty-two Wistar male rats were randomized into four groups: a control group (C) (no treatment), GM group (100 mg kg^−1^ bw day^−1^ GM), GM+SN group (100 mg kg^−1^ bw day^−1^ GM and 1 mL SN extract day^−1^), and SN group (1 mL SN extract day^−1^). Lipid peroxidation, evaluated by malondialdehyde (MDA), and antioxidant enzymes activity—superoxide dismutase (SOD), catalase (CAT), and glutathione peroxidase (GPX)—were recorded in renal tissue after ten days of experimental treatment. The MDA level was significantly higher in the GM group compared to the control group (*p* < 0.0001), and was significantly reduced by SN in the GM+SN group compared to the GM group (*p* = 0.021). SN extract failed to improve SOD, CAT, and GPX activity in the GM+SN group compared to the GM group (*p* > 0.05), and its action was most probably due to the ability of flavonoids (rutin, epicatechin) and ferulic and caffeic acids to inhibit synthesis and neutralize reactive species, to reduce the redox-active iron pool, and to inhibit lipid peroxidation. In this study, we propose an innovative method for counteracting GM nephrotoxicity with a high efficiency and low cost, but with the disadvantage of the multifactorial environmental variability of the content of SN extracts.

## 1. Introduction

Gentamicin (GM) is an aminoglycoside antibiotic with high efficacy in treating infections caused by Gram-negative bacteria, but with limited use due to its ototoxic and nephrotoxic effects. However, GM is still commonly used in pediatric clinical practice and the UK National Institute of Health and Care Excellence (NIHCE) guidelines recommend the use of GM (in combination with penicillin) as a first-line therapy in neonates with suspected early-onset sepsis [1]. On the other hand, the occurrence of carbapenemase-producing and/or colistin-resistant *Enterobacteriaceaes* requires the use of antibiotic combinations that include GM, a drug that was proven effective on these bacteria [2]. These aspects create the premise for increasing GM use and require solutions to counteract its ototoxic and nephrotoxic effects.

GM induces nephrotoxicity by concentrating in the proximal renal tubules, in the endosomal and lysosomal vacuoles [3], and in the Golgi complex [4], causing oxidative stress (OS), inflammatory and vascular responses, and finally, acute tubular necrosis. Several days after starting GM therapy, it may lead to non-oliguric renal failure, a slow rise in serum creatinine, and urinary hypo-osmolarity Renal OS is determined by the imbalance between reactive oxygen and nitrogen species (ROS, RNS) and antioxidant defense mechanisms (enzymatic and non-enzymatic).

In GM’s case, OS is caused by the increased production of reactive species (RS) and decreased enzymatic antioxidant defense, which both lead to the accumulation of free radicals and oxidative cascade reactions. In particular, due to the high lipid content, the kidney has an increased sensitivity to RS, leading to lipid peroxidation in the renal tissue [5].

To date, numerous studies have proven the nephroprotective effects of compounds with antioxidant and anti-inflammatory effects synthesized by plants and animals [6,7,8].

Of these natural compounds, flavonoids play an important role in prophylactic and curative intervention for both acute kidney injury (AKI) and chronic kidney disease (CKD) [9].

In a previously published study, we demonstrated nephroprotective effects, in terms of renal function and morphology, for *Sambucus nigra* L. elderflower extract rich in flavonoids, obtained from the wild specimens of European elder *Sambucus nigra* L. [10].

The elderflowers’ active principles consist of many polyphenolic compounds, including flavonoids such as kaempferol, quercetin, hyposide, astragalin, isoquercitrin, and rutin, chlorogenic acids and derivates, sterols, triterpene acids, free fatty acids, tannins, alkanes, mucilage, and sugar [11]. Previous studies have shown that *Sambucus nigra* L. elderflower infusion had an antioxidant activity higher than that of *Sambucus nigra* L. elderberry infusion [12], and the flower’s methanolic extract contained higher levels of phenolic compounds compared to the flower’s water extract [13]. The major phenolic constituents found in elderflower extracts were hydroxycinnamic acids (HCAs) and flavonol glycosides (Figure 1). Almost 75% of all the flavonols was represented by quercetin-3-rutinoside (rutin) [13], which has been proven to have beneficial effects in kidney pathology due to its antioxidant, anti-inflammatory, and antifibrotic activity [14,15].

The main HCAs isolated from elderflowers were represented by caffeic acid (CA) and its derivates, p-coumaric acid, and ferulic acid (FA) [17]. Previous studies have also demonstrated the nephroprotective effects of HCAs, due to their antioxidant and anti-inflammatory activity [18,19].

Because elderflowers have a high rutin content with proven antioxidant effects, the aim of the present was to check the possible protective effects of *Sambucus nigra* L. elderflower (SN) extract, obtained from elderflowers harvested in 2018, against lipid peroxidation in rats with experimentally GM-induced nephrotoxicity. In order to achieve this goal, we analyzed the effect of SN extract on the lipid peroxidation and on the activity of the main antioxidant enzymes in kidney tissue exposed to OS induced by GM. We concluded that SN extract inhibited lipid peroxidation, without improving enzymatic antioxidant activity in renal tissue.

In this study, we propose an innovative method of fighting against GM nephrotoxicity, with a high efficiency and low cost, but with the disadvantage of the multifactorial environmental variability of the content of SN extracts.

## 2. Materials and Methods

### 2.1. Collection and Processing of Elderflowers

To conduct the study, we used organic material represented by elderflowers (*Sambucus nigra* L. species) harvested in 2018 from the local spontaneous flora near Cluj-Napoca, Romania. *Sambucus nigra* L. inflorescences harvested without stalks were preserved by freezing at −18 °C in vacuum polyethylene packaging.

### 2.2. Quantitative Analysis of Phenolic Compounds in Elderflowers

The phenolic compounds, catechin, epicatechin, rutin (quercetin-3-*O*-rutinoside) and quercetin, vanillic acid, p-coumaric acid, and FA, were purchased from Sigma-Aldrich Co. (St. Louis, MO, USA). Analytical grade water was obtained from a Milli-Q Ultrapure water purification system (Millipore, Bedford, MA, USA). The formic acid and methanol were purchased from Merck (Darmstadt, Germany).

The preparation of samples for phenolic compounds analysis was done according to a modified version of the method proposed by Mikulic-Petkovsek et al. [20]. Ethanol was used as a solvent in which 1 g of crushed frozen SN sample was added and homogenized. The mixture was kept for 24 h at 2–3 °C, then the solvent was removed on a rotary vacuum evaporator (Laborota^®^ 4010 Digital, Heidolph, Schwabach, Germany). The resulting extract was recovered in 10 mL ethanol, filtered through a 0.45 μm Millipore^®^ filter, and kept frozen at −20 °C until analysis.

The phenolic compounds were assessed by the HPLC method described by Filip et al. [21]. Several studies have demonstrated that high-performance liquid chromatography (HPLC) is the most efficient technique for the qualitative and quantitative determination of polyphenolic compounds in plants, due to its high selectivity and precision [12,22,23,24,25,26,27,28].

Analyses were carried out on a Jasco Chromatograph (Jasco^®^ International Co., Ltd., Tokyo, Japan), equipped with an intelligent HPLC pump, an intelligent column thermostat, an intelligent UV/VIS detector, a ternary gradient unit, and an injection valve that was equipped with a 20 µL sample loop (Rheodyne^®^, Thermo Fisher Scientific, Waltham, MA, USA). Experimental data processing was performed with ChromPass^®^ software (version v1.7, Jasco International Co., Ltd., Tokyo, Japan).

Separation was carried out on a LiChrosorb^®^ RP-C18 column (25 × 0.46 cm) at 22 °C and UV detection at 270 nm. The mobile phase was a mixture of methanol (A, HPLC grade) and 0.1% formic acid solution (Millipore ultrapure water), and a gradient method was applied: 0–10 min, linear gradient 10–25% A; 10–25 min, linear gradient 25–30% A; 25–50 min, linear gradient 35–50% A; 50–70 min, isocratic 50% A, at the flow rate of 1 mL min^−1^. The injection volume was always 20 µL. The identification of the compounds was performed by comparing their elution times with those of the standard compounds (Sigma-Aldrich^®^), analyzed under the same HPLC conditions. The HPLC calibration curves were obtained with four concentration levels between 120 μg mL^−1^ and 11.25 μg mL^−1^, and the regression factors R^2^ were higher than 0.998 [21]. The recoveries were 98–101.2% and the relative standard deviations were ≤3.18% (*n* = 6). Limits of detection (LOD) and quantification (LOQ) were determined based on signal-to-noise, as follows: for catechin (1 and 3.3 µg mL^−1^), epicatechin (1.1 and 3.63 µg mL^−1^), vanillic acid (0.43 and 1.42 µg mL^−1^), caffeic acid (0.97 and 3.2 µg mL^−1^), p-coumaric acid (0.8 and 2.64 µg mL^−1^), ferulic acid (0.45 and 1.48 µg mL^−1^), rutin (0.38 and 1.26 µg mL^−1^), quercetin (1.1 and 3.63 µg mL^−1^), and luteolin (1.19 and 3.93 µg mL^−1^).

The stock solution of standards (1 mg mL^−1^ of each) was prepared in methanol and stored at 4 °C.

### 2.3. Protocol for Obtaining Elderflower Extract for In Vivo Administration

The “mother tincture” for in vivo administration was obtained from 100 g of finely ground frozen elderflower sample to which 1200 mL ethanol was added. After a rest of 24 h, a volume of 102.4 mL was separated to dryness with the vacuum rotary evaporator (Laborota^®^ 4010 Digital, Heidolph, Schwabach, Germany), the resulting residue being recovered in 32 mL of normal saline solution (sodium chloride, NaCl 0.9%), to obtain the SN extract for in vivo administration [29,30,31,32,33]. The amount of antioxidant compounds in the elderflower extract administered in vivo was calculated per 1 g of elderflower, according to the laws of proportionality (the rule of three).

### 2.4. Experimental Design

GM, a broad-spectrum aminoglycoside antibiotic, known for its nephrotoxic potential by oxidative mechanisms initiated in the proximal renal tubules where it generates ROS cascade, was used to induce nephrotoxicity.

The experiment was carried out at the University of Agricultural Science and Veterinary Medicine Cluj-Napoca, Romania, following Directive 2010/63/EU and national legislation (Law no. 43/2014). It was approved by the Committee for Bioethics and authorized by the State Veterinary Authority (No. 70/30.05.2017). The animals were caged at a controlled temperature (21–22 °C) and humidity (40–60%), with a 12/12 h light/dark cycle. Standard laboratory animal forage and water were freely available.

The study was conducted over ten days, in accordance with previous experimental models [34,35], and with the recommendations made by the medical guidelines for GM administration [36,37]. The experimental animals (32 male rats, adults from the Wistar line), with an average weight of 195 ± 10 g, were randomized into four groups, each composed of eight individuals. Rats in the control group (C) were injected with 1 mL of standard saline solution (0.9% NaCl) intraperitoneally daily and received 1 mL of normal saline solution (0.9% NaCl) by gavage daily. Rats from the GM group were injected with 100 mg kg^−1^ day^−1^ GM intraperitoneally daily and received 1 mL of normal saline solution (0.9% NaCl) by gavage daily. Rats from the GM + SN group were injected with 100 mg kg^−1^ bw day^−1^ GM intraperitoneally daily and received 1 mL of SN extract for in vivo administration (corresponding to 17.67 mg kg^−1^ bw rutin, 137.36 μg kg^−1^ bw CA, and 57.23 μg kg^−1^ bw FA) by gavage daily. Rats in the SN group received 1 mL of SN extract for in vivo administration by gavage daily and were injected with 1 mL of normal saline solution (0.9% NaCl) intraperitoneally daily.

On the 11th day, all animals were euthanized in the isoflurane euthanasia chamber.

After previous disinfection, the abdominal cavity was carefully dissected, and the right kidney was taken from each animal and frozen at −80 °C. After unfreezing, kidney samples were mechanically homogenized, separately, in the sterile 0.05 M phosphate-buffered saline (PBS) (pH = 7.5), and then the total amount of proteins was assessed by the Bradford method in each homogenate [38].

### 2.5. Analysis of Lipid Peroxidation

For quantifying lipid peroxidation, malondialdehyde (MDA) was assessed using 2-thiobarbituric acid by the fluorimetric method described by Conti, using a synchronous technique with excitation at 534 nm and emission at 548 nm [39]. The MDA values were expressed in nmol/mg protein.

### 2.6. Analysis of Antioxidant Enzyme Activity

#### 2.6.1. Superoxide Dismutase (SOD)

SOD activity was determined by using the cytochrome c reduction test, described by Flohe and Otting [40], into a mixture containing cytochrome c solution (2 μM in 50 mM phosphate buffer, pH 7.8), xanthine (5 μM), and cyanide (2 mM), in order to inhibit Cu and Zn SOD. Xanthine oxidase (0.2 U mL^−1^ in 0.1 mM ethylenediaminetetraacetic acid (EDTA)) was added to initiate the reaction. Increased absorbance at 550 nm, recorded for 5 min, indicated cytochrome c reduction. One unit of SOD activity is defined as the amount of enzyme able to inhibit by 50% the rate of cytochrome c reduction in specific conditions. Results were expressed in U/mg protein.

#### 2.6.2. Glutathione Peroxidase (GPX)

We measured GPX activity according to a slightly modified Flohe and Gunzler method by using a reaction mixture consisting of 10 mM glutathione (GSH), 2.4 U mL^−1^ glutathione reductase, and 1.5 mM NADPH, 1.5 mM H_2_O_2_ in 0.1 mM phosphate buffer (pH = 7.0) [41]. The reaction mixture was incubated at 37 °C, and the decrease in absorbance at 340 nm was recorded for 6 min.

The enzyme activity was defined as the amount of GPX that induces a GSH decrease by 10% from the initial concentration in one minute, at 37 °C and pH = 7, and was expressed in U/mg protein.

#### 2.6.3. Catalase (CAT)

According to the Pippenger method, we assayed CAT activity in the kidney based on absorbance modifications for a 10 mM H_2_O_2_ solution in 0.05 M potassium phosphate buffer (pH = 7.4) at 240 nm [42]. The enzyme quantity that produced a 0.43 reduction in absorbance per 3 min at 25 °C was defined as 1 unit (U) of CAT activity and was expressed as U/mg protein.

### 2.7. Statistical Analysis

All data were reported as mean ± SEM. The Gaussian distribution was checked by a D’Agostino and Pearson omnibus normality test. A one-way analysis of variance (ANOVA), followed by Bonferroni’s multiple comparison test procedure, was performed. Statistical significance was set at *p* < 0.05. Statistical values and graphs were obtained using GraphPad Prism version 5.0 for Windows (GraphPad^®^ Software, San Diego, CA, USA).

## 3. Results and Discussion

In the human body, ROS and RNS are normally synthesized in small quantities and play numerous roles in cell division and death, angiogenesis and vascular reactivity, adaptation to hypoxia and ischemia, antimicrobial defense, and hormonal synthesis. The best-known ROS are superoxide anion (O_2_^•−^), hydrogen peroxide (H_2_O_2_), and the hydroxyl radical (HO^•^). RNS are represented by nitric oxide (NO) and peroxy-nitrite (ONOO^−^), the latter being formed by the reaction of NO with O_2_^•−^ [43].

RS production is modulated by metabolic, hormonal, pro-inflammatory, and environmental factors [44,45,46,47], and, in normal circumstances, is quickly neutralized by enzymatic (mainly SOD, CAT, GPX), and non-enzymatic defense systems (GSH, melatonin, uric acid, bilirubin, vitamins C and E, plant polyphenols) [48].

OS appears when there is a hyper-production of RS or a reduction of antioxidant defense. OS causes lipid peroxidation, deoxyribonucleic acid (DNA) damage, protein denaturation, and damage to the mitochondria and cell membranes.

Due to its high content of long-chain polyunsaturated fatty acids targeted by ROS, the kidney is particularly vulnerable to OS and lipid peroxidation caused by nephrotoxic substances or drugs [5].

Previous studies demonstrated that GM induced renal OS by increasing RS synthesis and by reducing antioxidant defense. In this process, iron molecules take part in the ample and complex cascade of OS response to GM. After GM administration, RS are synthesized in the kidney, and iron is released by RS action and catalyzes further RS synthesis.

GM forms with iron the iron–GM complexes that further increase the synthesis of O_2_^•−^, H_2_O_2_, and HO^•^, the main ROS that cause OS [49].

On the other hand, redox active iron is able to initiate and perpetuate lipid peroxidation in cells [50]. Further H_2_O_2_ and renal lipid oxidation products can enhance ROS generation in mitochondria [51]. Additionally, ROS promotes intracellular iron increase by releasing iron from mitochondria, lysosomes, and ferritin [52].

In kidney tissue, ROS cause damage to the phospholipid membranes, proteins, DNA, mitochondria, and cytochrome c release, resulting in the necrosis or apoptosis of epithelial cells in the proximal renal tubules and vascular and mesangial changes [49,53,54,55,56].

Previous studies have shown that neutralizing HO^•^ radicals can reduce OS and GM’s nephrotoxic effects [49].

Several experimental studies assessed the effects of antioxidant compounds, and in 2018 Vargas reported that numerous flavonoids had nephroprotective effects, including against GM-induced nephrotoxicity [9], due to chemical structure and anti-oxidative activity.

The anti-oxidative activity of flavonoids is due to three main mechanisms: (1) the suppression of ROS synthesis by inhibiting synthesis enzymes or by chelating metal ions (iron, copper) involved in free radical generation; (2) the scavenging of ROS and RNS by donating hydrogen atoms to hydroxyl, peroxyl, and peroxynitrite radicals; and (3) the protection or upregulation of antioxidant defense.

Flavonoids protect lipids against oxidative damage by the same mechanisms involved in OS decreasing. Particularly, a 3′4′-catechol structure in the B ring of flavonoids enhances their capacity to inhibit lipid peroxidation. Rutin and epicatechin flavonoids have free-radical scavenging properties and powerfully inhibit lipid peroxidation [57].

Herbal extracts rich in rutin have been shown to have nephroprotective effects and have been intensively studied in recent years. To date, several studies have communicated the nephroprotective effect of rutin-rich extracts obtained from cactus cladodes (*Opuntia ficus-indica)* [58] and mulberry [59].

To our knowledge, the antioxidant effects of elderflower extract (also rich in rutin) on nephrotoxicity have not been studied to date. Moreover, elderflower extracts demonstrated their superiority to elderberry extracts in antioxidant potential assessed by the 2,2-diphenyl-1-picrylhydrazyl radical (DPPH) or ferric reducing antioxidant power (FRAP) assays, and confirmed the significant relationships obtained between the antioxidant properties and total phenolic and flavonoids [12,60].

### 3.1. Quantitative Evaluation of Phenolic Compounds in the Analyzed SN Ethanolic Extract

The main phenolic compounds identified in the analyzed SN ethanolic extract by HPLC-UV were catechin, epicatechin, rutin, CA, and FA (Table 1). Of these compounds, rutin recorded the highest concentration.

Figure 2 presents the HPLC-UV chromatograms of the standards mixture of phenolic compounds and the analyzed SN extract in ethanol.

As with other species, elderflowers may show variations in composition depending on the harvesting year, elderberry cultivar, exposure to pollutants, or electromagnetic fields [61,62,63,64]. Thus, it is particularly important to determine the composition of plant extracts used in each experimental model and to correlate the therapeutic effects with the main compounds found in the studied sample. The analyzed SN ethanolic extract had a concentration of 463.2 µg mL^−1^ rutin, 9 µg mL^−1^ epicatechin, 3.9 µg mL^−1^ catechin, 3.6 µg mL^−1^ CA, and 1.5 µg mL^−1^ FA, and other components in low concentrations.

A total of 1 g of elderflower was found in 10 mL of SN extract analyzed by HPLC and 1 mL of extract analyzed by HPLC contained 0.1 g of elderflower. The 1 mL of SN extract analyzed by HPLC contained 463.2 µg rutin, 9 µg epicatechin, 3.9 µg catechin, 3.6 µg CA, and 1.5 µg FA (see Table 1). Each experimental animal (average weight 195 g) received by gavage 1 mL of SN extract for in vivo administration, containing 0.744 g elderflower (a 134.4 mL final volume of SN extract for in vivo administration contained 100 g elderflower), which contained 3.44 mg rutin, 66.96 µg epicatechin, 29.01 µg catechin, 11.16 µg FA, and 26.78 µg CA, safe doses for in vivo administration. By converting the amounts to mg kg^−1^ bw, the result is that each experimental animal received 17.67 mg kg^−1^ bw rutin, 343.4 µg kg^−1^ bw epicatechin, 148.8 µg kg^−1^ bw catechin, 137.36 µg kg^−1^ bw CA, and 57.23 µg kg^−1^ bw FA.

In a previous study that used an elderflower extract with a composition similar to that used in the present study, we demonstrated the lack of nephrotoxicity and the improvement of renal function [10].

In another study, it was proven that cyanogenic glycosides, potentially toxic compounds, were present in low levels (adequate for human consumption) in the elderflowers collected at a foothill [65].

For rutin, the predicted median lethal dose (LD50) was calculated at >5000 mg kg^−1^ bw, corresponding to a toxicity class of five and indicating the safety of oral rutin administration [66]. The LD50 for CA was calculated at >100 mg kg^−1^ bw for birds [67], and intraperitoneally CA lower than 1250 mg kg^−1^ bw was non-lethal in rats [68]. FA has a low toxicity [69] and the LD50 was appreciated at 2445 mg kg^−1^ bw in male and 2113 mg kg^−1^ bw in female rats [70]. Epicatechin has an LD50 at 1000 mg kg^−1^ bw for intraperitoneal administration in mice [71]. In rats, the LD50 for oral administration of catechin was found at 2.428 mol kg^−1^ bw and oral rat chronic toxicity was 2.500 mg kg^−1^ bw day^−1^ [72]. All the LD50 communicated for the phenolic compounds identified in our study were much larger than the amounts to which the experimental animals were exposed.

### 3.2. Influence of SN Extract Administered In Vivo on Oxidative Stress Parameters

#### 3.2.1. In Vivo Influence of SN Extract on Lipid Peroxidation

In the kidney, the level of lipid peroxidation is used as a marker for the severity of kidney damage. Increased lipid peroxidation and the subsequent formation of advanced lipid peroxidation end products trigger pro-inflammatory pathways and activate the receptor for advanced glycation end products, with the subsequent activation of ROS and cytokine production [51]. MDA is the main marker of lipid peroxidation. Its presence confirms the presence of tissue OS. Previous studies have shown that an increased MDA level in the renal tissue, induced by GM, was associated with morphological and functional impairment [73,74,75,76]. Consistent with these results, in our study GM administration led to a considerable increase in lipid peroxidation.

The MDA level was significantly higher in the GM group compared to the control (C) group (*p* < 0.0001). In the GM+SN group, the SN extract for in vivo administration prevented GM-induced MDA growth. Thus, the MDA level was 1.6 nmol/mg protein in the GM group, compared to 1.1 nmol/mg protein in the GM+SN group and 0.5 nmol/mg protein in the C group. The reduction of the MDA level was 34.9% in the GM+SN group compared to the GM group, which is statistically significant (*p* = 0.021). Nevertheless, for the protocol applied in this experimental model, the SN extract for in vivo administration did not completely eliminate the effects of GM. Thus, the MDA level was significantly higher in the GM+SN group than in the C group (*p* = 0.028) (Figure 3).

In this study, we demonstrated for the first time the effect of SN extract for in vivo administration to significantly reduce lipid peroxidation, induced by experimental exposure to GM, in the renal tissues.

In GM-induced nephropathy, numerous plant extracts and antioxidant compounds have been found to decrease lipid peroxidation and to provide nephroprotection. Some of them have been analyzed by Casanova in a systematic analysis. In this analysis, calcium dobesilate and *Zingiber o**fficinale* (ginger) extract showed the best nephroprotective profile associated with antioxidant activity. Antioxidant and nephroprotective activity was also proven for melatonin, erdosteine, rosiglitazone, garlic, S-allylcysteine, b-sitosterol, L-carnitine, D-carnitine curcumin, *Spirulina platensis*, soybean extract, grape seed extract, olive leaf extract, *Zingiber officinale* extract, *Sida rhomboidea* leaf extract, *Sonchus asper* extract, *Morchella esculenta* mycelium extract, and *Nigella sativa* oil. Arabic gum, carvedilol, and rosmarinic acid failed in nephroprotection despite their antioxidant activity [6]. Apart from the compounds analyzed by Casanova, the MDA level was also alleviated in GM-induced nephrotoxicity experimental models by *Zataria multiflora* hydroalcoholic extract [77], *Pimpinella anisum* L. ethanolic extract [78], *Pistacia atlantica* leaf hydroethanolic extract [79], pomegranate extract [80], *Helichrysum plicatum DC.* subsp. *plicatum* extract [81], *Ginkgo biloba* extract [82], grape seed extract [83], crude extract and solvent fractions of *Euclea divinorum* leaves [84], beet root (*Beta vulgaris* L.) ethanolic extract [85], aqueous extract of root of *Boerhavia diffusa* [86], *Crocus sativus* (known as saffron) [87], aqueous garlic extract [88], Riceberry bran extract [89], *Tephrosia purpurea* (L.) *Pers.* leaves ethanolic extract [90], hydroalcoholic *Malva sylvestris* extract [74], *Urtica dioica* methanolic leaf extract [91], ethyl acetate fraction from *Rotula aquatic* [92], *Origanum vulgare* L. extract [93], wild bilberries (*Vaccinium myrtillus* L.) extract [94], and ginger (*Zingiber officinale*) and turmeric (*Curcuma longa*) rhizomes administrated as dietary supplements [95]. The antioxidant activity of elderflower extract was compared to that of elderberry and elder leaves and was found to be highest in flowers [16,59]. Elderflower extract also showed the highest inhibition of DPPH in comparison with rutin, neutralized reactive hydroxyl radicals more effectively than rutin and quercetin, and showed metal-chelative properties stronger than those of rutin and quercetin [96,97].

In these circumstances, we believe that a flavonoid-rich SN extract for in vivo administration, which acts by reducing MDA in renal parenchyma, could be considered a useful nephroprotective agent for combating histological lesions caused by GM.

#### 3.2.2. In Vivo Influence of SN Extract on Superoxide Dismutase, Catalase, and Glutathione Peroxidase

GM induces synthesis of an increased amount of RS, both ROS (O_2_^•−^, H_2_O_2_ and OH^•^) and RNS (NO and ONOO^−^), in the renal parenchyma [98].

In the kidney, RS have different effects on inducible antioxidant enzymes, such as SOD, CAT, and GPX, depending on their amount and action period [99,100,101,102]. Thus, in short-term stimulation, RS can induce the synthesis of antioxidant enzymes, confirmed by the studies published by Medić and Martínez-Salgado that showed increased SOD activity after exposing rats to GM for 7 days and exposing renal mesangial cells to GM for 8 h [103,104], respectively. Dobashi et al. reported an increase in GPX gene expression after the exposure of rat kidney cells (KNRK) to a NO donor [102]. On the other hand, a long exposure (8–10 days) led to exhaustion of the synthesis of these enzymes, as demonstrated by studies in which exposure to GM decreased SOD, CAT, and GPX activity [105,106].

Some studies have shown statistically insignificant changes in SOD and CAT after exposure to GM [103,107,108]. Thus, it is even more difficult to interpret the seemingly contradictory results of the studies in the literature.

Consistent with previously published data reporting reduced SOD and GPX activity as an effect of GM [108], in our study GM administration produced a significant decrease in SOD and GPX activity in the renal tissue in the group with GM compared to the control group.

Our results show that the SOD level decreased significantly (*p* < 0.01) by 19.3% in the GM group compared to the untreated C group. In the GM+SN treated group, the renal SOD level was reduced by only 9.8% compared to the C group. SN extract for in vivo administration treatment did not significantly increase the renal SOD level after the induction of nephrotoxicity with GM: the renal SOD level increased by 23.2% in the GM+SN group compared to the GM group (*p* > 0.05). The SOD level was 13.3% higher in the SN group than in the C group (*p* > 0.05) (Figure 4).

The CAT level recorded no statistically significant difference between groups.

The CAT value was not significantly influenced by the administration of SN extract in the GM+SN group compared to the GM group (*p* = 0.7367) or in the SN group compared to the C group (*p* = 0.9889), respectively (Figure 5).

Concerning GPX, its level was significantly reduced (by 29.1%) in the GM group compared to the C group (*p* < 0.0001). GPX was increased in the SN group compared to the C group, but the increase was not statistically significant (*p* = 0.5622) (Figure 6). Similar to the SOD activity, the GPX activity increased significantly only when comparing the SN group with the GM+SN group (<0.0001) and with the GM group (*p* < 0.0001) (Figure 4 and Figure 6).

SOD is the first defense line against O_2_^•−^. Its main role is to prevent the transformation of O_2_^•−^ into ONOO^−^ and HO^•^, highly reactive compounds that fragment DNA and membrane lipid structures.

SOD deficiency allows the combination of O_2_^•−^ with NO, with two major negative consequences. On the one hand, NO loses the function of a signaling molecule and with it the beneficial effects of regulating blood flow in the renal cortex, and on the other hand generates ONOO^−^, an extremely aggressive compound for renal structures.

ONOO^−^ disrupts the mitochondrial respiratory chain and induces lipid peroxidation, protein nitrosylation, enzymatic inhibition [109], and the apoptosis of renal cells [110].

In the presence of free ions of iron (Fe^+2^) and copper (Cu^+2^), another harmful compound, O_2_^•−^, which is not neutralized by SOD, can generate HO^•−^ radicals, fragmenting DNA, proteins, and lipids [111].

SOD activity must be coordinated with that of CAT and GPX, participating in the neutralization of H_2_O_2_ generated by SOD action.

CAT intervenes to remove H_2_O_2_ by reactions resulting in harmless molecules of water.

GPX is an enzyme with a major role in the antioxidant defense in kidneys, as it is able to participate in the neutralization of both H_2_O_2_ and organic peroxides [112]. GPX is considered an enzyme with strong activity in neutralizing fatty acid hydroperoxides [113].

GPX deficiency allows H_2_O_2_ to diffuse into tissues and act as an RS. The main mechanism by which H_2_O_2_ acts at the renal level is the inhibition of pexophagy, which allows for the structural recovery of peroxisomes and ROS generation limitation at the peroxisomal level [114]. In the renal cortex, H_2_O_2_ mediates iron release from the mitochondria by GM [115].

In our study, a less expected result was in the CAT level changes. In the GM exposed group, CAT was lower (but not significantly) than in the control group, and after the administration of SN, CAT did not record a significant increase in the GM+SN group compared to the GM group. These results can be explained by the fact that changes in CAT expression are influenced by the type of analyzed tissue and its basal enzymatical antioxidant capacity [116].

Another unexpected result of our study was that SN failed to restore the enzymatic activity of GPX and SOD. These had only a statistically insignificant increase in the GM+SN group compared to the GM group.

In a previous study, Su reported an increase in GPX activity after rutin administration in mice exposed to intense physical exertion [117]. Moreover, rutin administered in rats at a dose of 150 mg kg^−1^ for 8 days restored GPX activity, reduced after 80 mg kg^−1^ bw day^−1^ GM, simultaneously with the reduction of OS, the improvement of renal function, and the reduction of histological lesions [118]. In our study, GPX recorded only an insignificant increase after the administration of SN. This result can be explained by the fact that the GM dose was higher and the rutin amount in SN was lower than those in the abovementioned studies.

The failure of GPX increase could also be explained by the presence of compounds that interfere with antioxidant enzymes activity. Simos has shown that the flavanols catechin and epicatechin, also present in our elderflower extract, could reduce GPX activity [119].

In a previous GM-induced experimental model of nephrotoxicity in rats, pre-treatment (6 days) and concomitant treatment (8 days) with 150 mg kg^−1^ of rutin attenuated nephrotoxicity induced by 80 mg kg^−1^ GM by increasing SOD, CAT, and GPX activity [118].

In our study, the amount of rutin administered to each animal was much smaller (17.67 mg kg^−1^) and administration was done concomitantly with GM, which allowed some amount of SOD to be consumed in redox reactions. Another explanation could be a differentiated stimulation of SOD isoform synthesis.

The kidney tissue contains three isoforms of SOD. SOD1 and SOD3 are located in the cytosol and the extracellular space, respectively. SOD2 (Mn-SOD) is found in the mitochondria and plays a vital role in keeping O_2_^•−^ at low levels in the mitochondrial matrix by conversion to H_2_O_2_ [120,121].

The SOD1 isoform accounts for up to 80% of the total SOD activity in the mammalian kidney [122]. SOD2 ablation results in a more severe pathological phenotype than SOD1 ablation by worsening OS [51].

In a previous study, Sitarek demonstrated that *Leonurus sibiricus* L. plant extract, which contains catechin, quercetin, rutin, CA, and FA, increased encoding SOD2 gene expression in Chinese hamster ovary cells exposed to H_2_O_2_ [123]. These five compounds were also found in the elderflower extract we studied. Therefore, we believe that the studied SN especially stimulates the synthesis of SOD2, essential for reducing OS at the mitochondrial level and for reducing lipid peroxidation, without generating a statistically significant increase in the level of total SOD activity.

### 3.3. Non-Enzymatic Antioxidant Effects of Bioflavonoids from Elderflower Extract and Their Main Compound—Rutin

The main compound from the analyzed SN extract in our study was rutin, a bioflavonoid, glycoside of quercetin, with antioxidant and anti-inflammatory effects [118]. The mechanisms by which rutin reduces tissue OS are correlated with its hydrogen donor capacity, which confers on it the function of scavenger for ROS and RNS, and with the chelating property of metals that confers the ability to neutralize iron ions and to form inactive, stabile iron compounds [117].

GM causes an increase in mitochondrial H_2_O_2_ synthesis at the renal level in a dose dependent manner [124] and induces a H_2_O_2_-mediated iron mobilization from mitochondria [115]. In an environment with H_2_O_2_, free iron catalyzes the production of high levels of ROS: O_2_^•−^, HO^•^, and OH^–^ by the Haber–Weiss reaction, initiated by the Fenton reaction [125].

Normally, most cellular iron is safely stored in organelles—lysosomes, mitochondria—or in ferritin, but in chronic or acute stress, a labile iron pool (defined as a chelatable and redox-active pool of iron, comprising both ionic forms of iron: Fe^2+^ and Fe^3+^) increases [126]. Free iron is a key player in the Haber–Weiss reaction, using H_2_O_2_ and O_2_^·−^ to produce more toxic HO^•^ radicals and catalyzing lipid peroxidation and the oxidative damage of proteins that are strongly associated with kidney disease and its severity [51,127].

In fact, iron, similar to ROS, functions as an inducible stress intracellular messenger that serves to regulate cell death responses. Iron chelators can act as powerful antioxidants [128]; Eid et al. communicated in 2017 that intracellular iron chelators were found to function as ROS scavengers [52].

It was proven that elderflower extract has metal-chelative properties stronger than those of rutin and quercetin standards, their main effect being to prevent the initiation of HO^•^ radicals synthesis [100,129].

If we accept that elderflower extract protects the kidney against gentamicin-induced OS by chelating iron ions, and if we take into account that the chelating properties of the whole elderflower extract are superior to those of rutin and quercetin standards, we can consider that elder flower extract with its full polyphenol content is preferable for controlling renal OS to the detriment of standardized extracts that use only one type of polyphenols.

Flavonoids can act as antioxidants per se, independently from constitutively enzymatic or non-enzymatic antioxidant cell systems, and could have an additive effect to the endogenous scavenging compounds [130].

In a previous study, Robak demonstrated that many flavonoids inhibited lipid peroxidation, mainly via the scavenging of O_2_^•−^ anions, whereas other non-flavonoid antioxidant compounds act on free radical reactions via HO^•^ radicals scavenging [131].

Rutin inhibits lipid peroxidation, working as an ROS scavenger by donating hydrogen atoms to all RS involved in the peroxidation of lipids: O_2_^•−^ anions, singlet oxygen, HO^•^ radicals, and peroxy radicals. Peroxyl radicals scavenging by rutin is essential for preventing lipid peroxidation by interrupting the propagation of free radical chain reactions [132]. Rutin also works as fast-acting antioxidant by its ability to intervene as a terminator for lipid peroxidation by chelating metal ions [29].

In GM administration, rutin most likely intervenes from the initial stages of OS generation by blocking the iron ions released from mitochondria and preventing the formation of pro-oxidant iron–GM complexes. Consequently, ROS (O_2_^•−^, H_2_O_2_, and OH^•^) generating cascade reactions are considerably limited. Additionally, by its ROS scavenger function, rutin neutralizes H_2_O_2_ released by GM from mitochondria, causing a blockade of H_2_O_2_-dependent iron release and a further reduction in the OS, lipid peroxidation, and MDA synthesis induced by redox-active iron.

In our study, rutin’s action of neutralizing ROS, chelating iron ions, and blocking GM–iron complexes formation most likely represents the main mechanism that explains the reduction in MDA levels, without antioxidant enzymes increase, in the group of rats that received GM and SN extract for in vivo administration with a high rutin content.

We also believe that co-occurring flavonoids work synergistically with rutin to antagonize ROS and to inhibit lipid peroxidation.

Together, rutin, epicatechin, and catechin have been proved to have a HO^•^ scavenging capacity 100–300 times higher than that of mannitol, a typical HO^•^ scavenger. Moreover, all flavonoids, except epicatechin, had an inhibitory effect on xanthine oxidase involved in the synthesis of O_2_^•−^ anions [133].

Another property of rutin involved in lipid peroxidation decrease is its ability to inhibit NO production. The involvement of NO in the initiation of lipid peroxidation was reported for all situations when it acts in the presence of O_2_^•−^ anions to form peroxynitrite, a powerful oxidant able to initiate lipid peroxidation. To inhibit NO production, rutin acts together with CA, another strong inhibitor of NO synthesis [134].

### 3.4. Role of Caffeic Acid and Ferulic Acid

A positive correlation between CA derivatives and FA derivatives and antioxidant activity was demonstrated in many studies [135,136].

CA is a compound that belongs to the HCAs class and can chelate ferrous ions (Fe^2+^) [137]. Fe^2+^ accelerates lipid oxidation by breaking down H_2_O_2_ and lipid peroxides to reactive free radicals via the Fenton reaction [138].

Due to the chelating properties of Fe^2+^, CA acts as a compound with an inhibitory effect on lipid peroxidation and MDA synthesis.

Another recognized property of CA involved in lipid peroxidation reduction is the ability to inhibit NO synthesis [134].

FA present in SN extract for in vivo administration is a phenolic compound that possesses a free radical scavenging capacity. The carboxylic acid group acts as an anchor for the lipid bilayer and confers FA properties against lipid peroxidation. Because FA scavenges O_2_^•−^ and inhibits lipid peroxidation induced by O_2_^•−^, FA activity is considered similar to that of SOD [139].

Tannic acid is another natural polyphenol with antioxidant, antimicrobial, and anti-inflammatory properties, but is also able to influence the structure and conformation of fibrinogen to form fibrin. In comparison with tannic acid, which could potentially be dangerous for renal fibrosis development and the poor evolution of kidney diseases [140,141,142], in elderflower extract, antifibrotic and nephroprotective effects against gentamicin-induced nephrotoxicity have been proven [10].

### 3.5. New Research Direction for Elderflowers Extract

Our study is far from completely presenting and explaining all the therapeutic effects of elderflower extract.

#### 3.5.1. Action Mechanism of Active Compounds

In the future, the antioxidant mechanisms for rutin, epicatechin, catechin, FA, and CA, administered alone or in combination, must be evaluated in experimental models of nephrotoxicity to gentamicin. The ability of phenolic compounds to scavenge free radicals must be evaluated using the 2,2′-azino-bis(3-ethylbenzothiazoline-6-sulfonic acid) (ABTS^•+^) radical cation-based test (for cation radicals) or 2,2-diphenyl-1-picrylhydrazyl (DPPH) radical-based test (for stable radicals). For the antioxidant activity assay, a wide variety of methods may be used, including chemical-based methods such as the cupric ions reducing power assay or FRAP and biological assays, such as cellular antioxidant activity. The peroxyl radical, superoxide radical anion, hydrogen peroxide, hydroxyl radical scavenging assay, and singlet oxygen quenching assay must by performed for specific antioxidant activity [143]. The near hydrogen atom transfer mechanism, sequential proton loss–electron transfer, and single electron transfer–proton transfer must be also studied with density functional theory methods [144]. For finding a relationship between structural features of compounds and their activities, the development of a quantitative structure–activity relationship model should be considered [145].

Another question logically arising is whether active compounds act synergically. In a previous study, it was demonstrated that rutin pretreatment could attenuate GM-induced nephrotoxicity by increasing GSH levels and SOD, CAT, and GPx activity, as well as by reducing MDA level [118]. In an experimental model, catechin prevented a GM-induced decrease in renal GSH, concomitantly improving renal function [146]. Epicatechin was not studied in GM-induced nephrotoxicity, but it was found to alleviate mitochondrial structural changes and ROS level caused by cisplatin in the renal cortex of mice [147], alterations observed also in GM-induced nephrotoxicity [148]. Beneficial effects in kidney injury caused by GM have been also proven for HCA. Pretreatment with caffeic acid phenethyl ester was effective in preventing the rise of MDA level induced by GM [149], and FA reduced MDA level and OS by increasing SOD activity in GM-induced nephrotoxicity [150]. In another study, phenolic natural compounds presented better antioxidant capacities when they were tested in a mixture compared to the antioxidant activities displayed individually by each of them in a GM nephrotoxicity experimental model, suggesting synergistic antioxidant effects [151]. Another study showed that the extract of *Sambucus nigra* L. elderflowers had metal-chelative properties stronger than those of isolated rutin and quercetin standards, preventing the initiation of the synthesis of hydroxyl radicals. A similar superiority of *Sambucus nigra* L. elderflower extract compared to rutin and quercetin was found for radical scavenging and the elimination of hydroxyl radicals [96]. In our study, all phenolic antioxidant compounds were administered at lower amounts compared to those in the previous studies, leading us to consider that the synergistic effect of these compounds merits study in the future.

Other research directions that deserve to be developed refer to the effects of elderflower extract on NO and inducible nitric oxide synthase (iNOS), caspase-3, and light chain 3B (LC3B), because previous studies demonstrated that rutin pre-treatment attenuated nephrotoxicity induced by OS after GM administration by inhibiting iNOS, cleaving caspase-3, and LC3B [118].

#### 3.5.2. Interference between SN Extract and Antimicrobial Effect of GM

Even if mechanisms for antimicrobial effects of GM are not fully elucidated, some authors have suggested that oxidation reduction reactions are involved in bacterial death [152]. In this context, a logical question to be raised is whether the combination of GM with antioxidants is beneficial for the antibacterial effect.

Contradictory results are communicated for the association between GM and antioxidants. The simultaneous administration of GM and luteolin enhanced the antibacterial activity of GM against *S. aureus* and *E. coli*. [153]. Quercetin, another antioxidant, did not substantially modify the antibacterial activity of gentamicin against *E. coli* and contributed to the enhancement of GM activity against *S. aureus*. In a previous study, Miroshnichenko demonstrated that other antioxidants, ascorbic acid, methylethylpyridinol, and N-acetylcysteine, reduced the activity of GM in vitro and in vivo without decreasing nephrotoxic effects [154]. On the other hand, antioxidant and antimicrobial activities have been demonstrated for many natural compounds from plants. Many of these compounds are phenols found in medicinal plants (*Curcuma longa* L., *Kalanchoe delagoensis* L., *Asparagus aethiopicus* L., *Senna alexandrina* L., *Citrullus colocynthis* L., *Gasteria pillansii* L., *Brassica juncea* and *Cymbopogon citratus*, L. *glaucescens*, *Convolvulus austro-aegyptiacus*, and *Convolvulus pilosellifolius*) and in marine plants (*Ecklonia cava*) [155]. For many culinary herbs rich in flavonoids, concomitant antimicrobial and antioxidant properties, useful in food preservation, have been proven [156]. Krawitz demonstrated that standardized elderberry extract possesses antimicrobial activity against both Gram-negative bacteria (*Branhamella catarrhalis*) and Gram-positive bacteria (*Streptococcus pyogenes* and group C and G *Streptococci*) [157].

In a previous study, it was proven that elderflower extract had an inhibitory activity against a wide range of nosocomial pathogens, namely Gram-positive (*Staphylococcus* sp., *B. cereus*) and Gram-negative (*Salmonella* Poona, *P. aeruginosa*) pathogens, and the highest inhibitory activity towards methicillin-resistant *Staphylococcus aureus* (MRSA) [158]. To our knowledge, no studies have been conducted to date on GM and elderflower extract association. Because the inhibitory or additive/stimulative effects of the association between GM and elderflower extract are dependent on the concentration of the partners, the bacteria and host organism, this aspect must be studied in the future.

#### 3.5.3. Pharmacological Characteristics of SN Extract

The EC50 is the concentration of a drug that gives a half-maximal response and is a fundamental concept in pharmacology. Because SN extract’s composition may differ due to environmental factors, it is important to find the EC50 for the whole extract and for each active antioxidant compound, and to verify if the EC50 is reached in all normal environmental conditions (temperature, altitude, rainfall, sunlight). These findings will contribute to the optimal use of elderflower in the food and pharmacological industry.

In order to improve efficiency, curves for time-dependent activity of the whole SN extract should also be calculated in the future.

At the same time, it is worth studying *Sambucus nigra* effects in other conditions where OS is an important pathogenetic mechanism, e.g., in metabolic, cardiovascular, and neurologic diseases.

#### 3.5.4. Efficiency of SN Extract in Other Nephrotoxicity Models

The nephroprotective effects of elderberry should also be studied as they relate to the use of other nephrotoxic compounds, such as alcohol, non-steroidal anti-inflammatory drugs (NSAIDS), chemotherapeutics, immunosuppressants, and contrast agents [159,160,161].

While effective alternatives have been proposed for the use of NSAIDS, unfortunately this is not the case for chemotherapeutics, immunosuppressants, and contrast agents [162,163,164].

#### 3.5.5. Antiviral Effect in SARS-CoV-2 Infection

In the context of the SARS-CoV-2 pandemic, elderberry fruit extract has been discussed as a potential antiviral drug [165], but also as a possible enhancer of the inflammatory reaction [166]. In a previous study, we demonstrated the renal antifibrotic effect of elder flower extract (*Sambucus nigra* L. species) [10], with a higher bioflavonoid content than elderberry fruits. On the other hand, OS, which can be reduced by elderflower extract, are one of the main mechanisms involved in pulmonary fibrosis secondary to SARS-CoV-2 infection. Considering these aspects, it is worth studying the type of elder plant extract and the optimal time of administration in SARS-CoV-2 infection.

### 3.6. Elderflowers—Promising Medical and Nutritional Intervention in Kidney Diseases

For a long time, elderflowers and berries have been widely used in nutrition and traditional therapy, and more recently in the food industry. As natural flavoring components, they can be found in alcoholic and nonalcoholic beverages, sparkling, bitter, and white wine, fruit brandies, and various spirits, as well as in tea and products such as yoghurt or ice cream [129].

Elderflowers can be used both as standardized extracts and as functional foods. Moreover, *Sambucus nigra* L. flowers contain even higher amounts of phenolic compounds than elderberries and leaves [16], and thus usually also have a higher antioxidant activity. As both characteristics are maintained at thermal processing, elderflowers can be widely used in food and pharmaceutical industries [129].

Our study validated the hypothesis that SN extract rich in polyphenolic compounds may provide nephroprotection in GM-induced nephrotoxicity, due to the antioxidant action. This characteristic recommends it to be studied further, in order to use it for medical purposes and in the food industry.

### 3.7. Limits of the Study

Our study demonstrated only the ability of elderberry flower extract to inhibit GM-induced lipid peroxidation, without demonstrating its action mechanism. Flavonoid-rich content was proved for the elderflower extract, but the action mechanism was only taken from other studies and not demonstrated for our experimental model. Another limit of this study was the multifactorial environmentally induced variability of elderflowers composition, which means that the results of the present study can only be extrapolated to similar extracts.

## 4. Conclusions

In our experimental model, elderflower extract rich in rutin, CA, and FA significantly reduced GM-induced lipid peroxidation, quantified by MDA levels, in the renal tissue.

Because we did not observe a statistically significant increase in the levels of the antioxidant enzymes SOD, CAT, and GPX, we are entitled to believe that the antioxidant effect of elderflower extract was due to the reduced production of RS and the direct inhibition of lipid peroxidation by rutin, epicatechin, FA, and CA, found in significant quantities in the SN extract. Rutin and CA, which have the property of chelating iron ions and reducing labile, redox-active pools of iron, were probably the main factors responsible for the decrease of RS synthesis after the iron release from mitochondria, lysosomes, and protein complexes caused by renal exposure to GM.

In this context, we believe that elderflower extract could be considered a useful nephroprotective agent against lipoperoxidation caused by GM.

## Figures and Tables

**Figure 1 materials-15-00772-f001:**
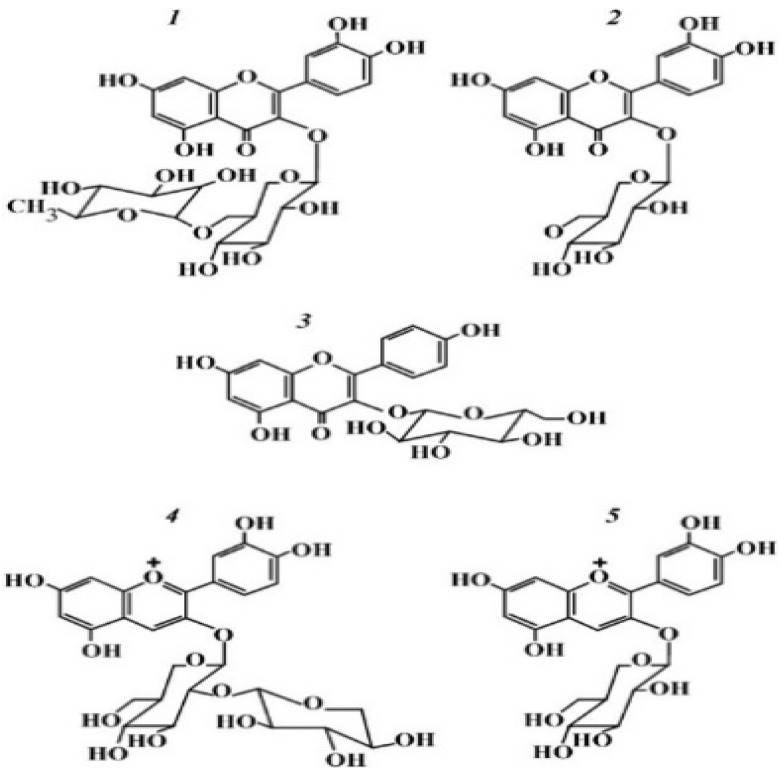
Chemical structures of the main flavonols from *Sambucus nigra:* rutin (**1**); isoquercitrin (**2**); astragaline (**3**) and antocyanins: cyanidin-3-sambubioside (**4**); cyanidin-3-glucoside (**5**) (Created with BioRender.com) [16].

**Figure 2 materials-15-00772-f002:**
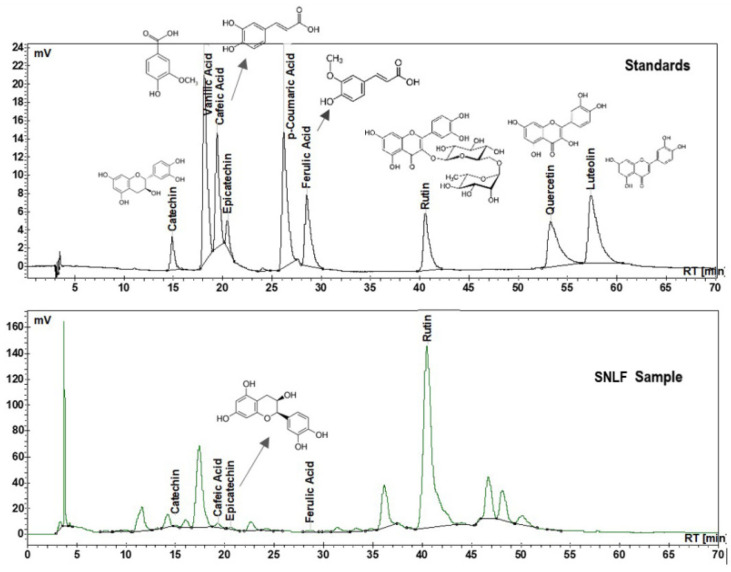
HPLC-UV chromatograms of standards mixture of studied phenolic compounds and analyzed SN ethanolic extract.

**Figure 3 materials-15-00772-f003:**
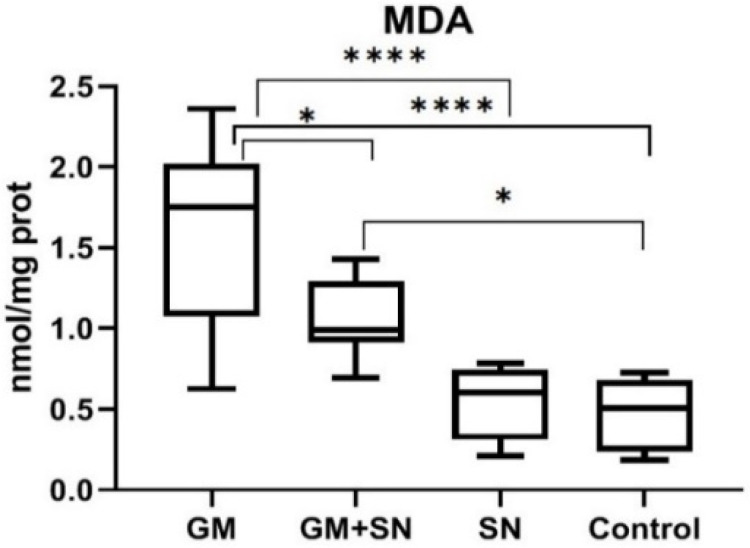
Malondialdehyde (MDA) level in the investigated groups (* *p* < 0.05, **** *p* < 0.0001). GM—gentamicin group, GM+SN—gentamicin and *Sambucus nigra* group, SN—*Sambucus nigra* group, Control—control group (no treatment).

**Figure 4 materials-15-00772-f004:**
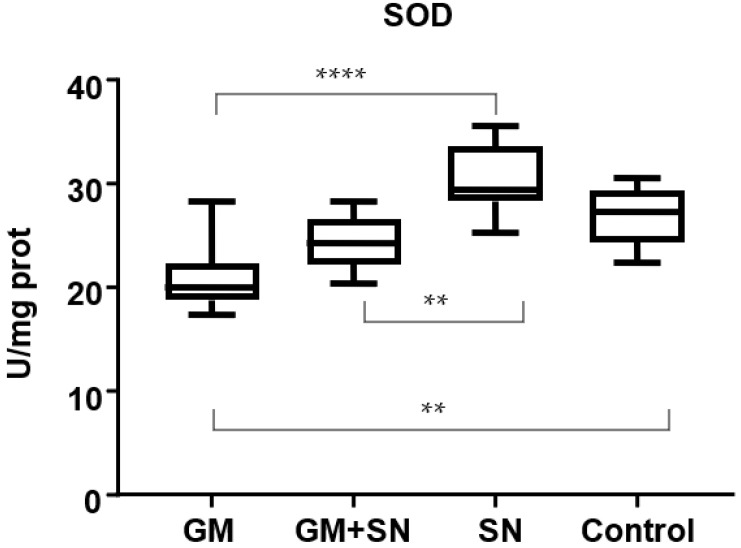
Superoxide dismutase (SOD) level in the investigated groups (** *p* < 0.01, **** *p* < 0.0001). GM—gentamicin group, GM+SN—gentamicin and *Sambucus nigra* group, SN—*Sambucus nigra* group, Control—control group (no treatment).

**Figure 5 materials-15-00772-f005:**
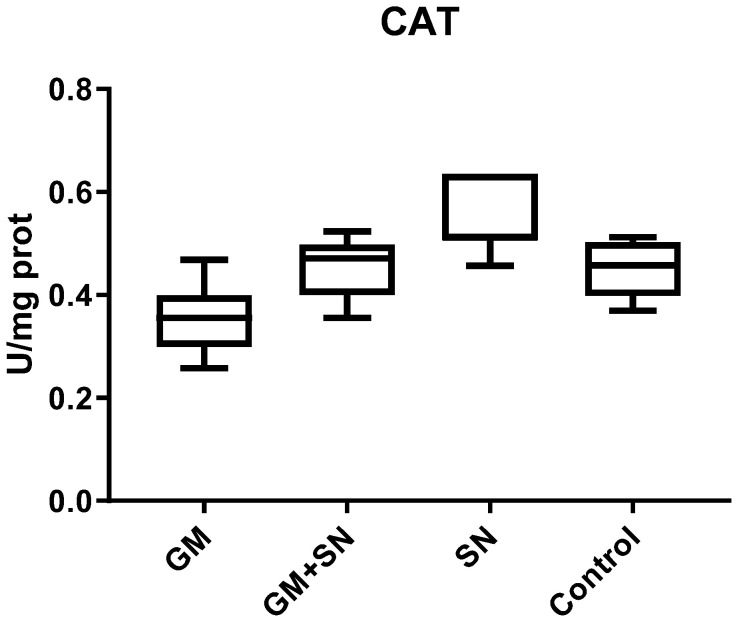
Catalase (CAT) level in the investigated groups. GM—gentamicin group, GM+SN—gentamicin and *Sambucus nigra* group, SN—*Sambucus nigra* group, Control—control group (no treatment).

**Figure 6 materials-15-00772-f006:**
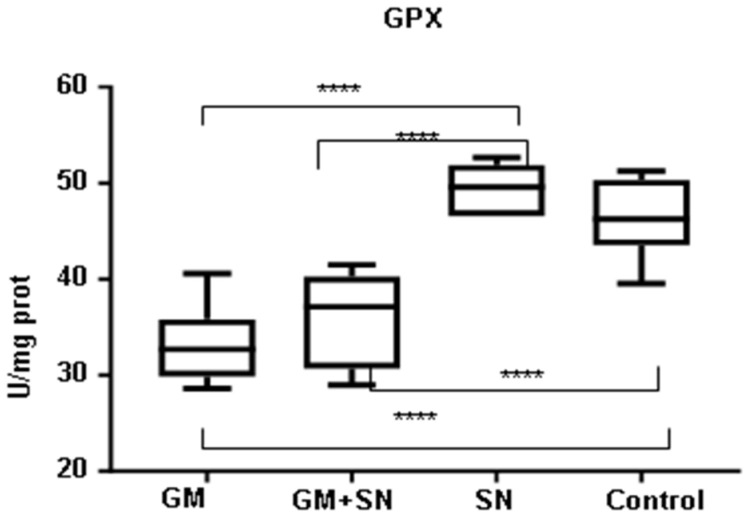
Glutathione peroxidase (GPX) level in the investigated groups (**** *p* < 0.0001). GM—gentamicin group, GM+SN—gentamicin and *Sambucus nigra* group, SN—*Sambucus nigra* group, Control—control group (no treatment).

**Table 1 materials-15-00772-t001:** The main phenolic compounds identified by HPLC-UV in the analyzed SN ethanolic extract.

Polyphenolic Compounds	Amount of Polyphenolic Compounds (μg mL^−1^)	
	X ^1^ ± St. Dev. ^2^	±St. Dev. ^2^
**Flavanols**		
Catechin	3.9 ± 0.3	
Epicatechin	9.0 ± 1.1	
**Flavonols**		
Quercetin-3-*O*-rutinoside (rutin)	463.2 ± 0.0	
**Hydroxycinnamic acids**		
Caffeic acid	3.6 ± 0.1	
Ferulic acid	1.5 ± 0.3	

^1^ X —mean value, ^2^ St. Dev.—standard deviation.

## Data Availability

The data presented in this study are openly available in FigShare at https://doi.org/10.6084/m9.figshare.18515198.

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
