# Peer review of "A Flavonoid-Rich Extract of Sambucus nigra L. Reduced Lipid Peroxidation in a Rat Experimental Model of Gentamicin Nephrotoxicity"

_materials, 2022, doi:10.3390/ma15030772_

Round 1

Reviewer 1 Report

  1. As described by the authors, SNLF composed of rutin, epicatechin, ferulic and caffeic acid. When specific experimental results are obtained, how can the functions performed by specific components be determined?

  1. There are some formatting errors in the article. For example, space is required between the number and the character. Kindly check for correctness.

  1. Notes are required for the first appearance of the abbreviation, unless it is something well known such as DNA, RNA, etc. Abstract: SN. Please check carefully and use it properly.

  1. What are the advantages of the SNLF in this paper compared to other classic Antioxidants (like tannic acid, 10.1016/j.ijbiomac.2021.12.007)? The comparison and the advantages can be added to the manuscript.

  1. Spelling of references must be checked to meet the journal style.

Author Response

  1. As described by the authors, SNLF composed of rutin, epicatechin, ferulic and caffeic acid. When specific experimental results are obtained, how can the functions performed by specific components be determined?

Response 1: We recognized, as one of the study limits, that the specific actions of each active compound (rutin, epicatechin, ferulic and caffeic acid) were not studied in our experimental model. These specific actions recorded in the literature have been identified and cited as mechanisms involved in the favorable effects of elderflower extract.

We mentioned as future research directions the mechanisms of action for rutin, epicatechin, catechin, ferulic and caffeic acid, administered alone or in combinations, in experimental models of gentamicin nephrotoxicity.

2. There are some formatting errors in the article. For example, space is required between the number and the character. Kindly check for correctness.

Response 2: We checked the manuscript for formatting errors. We identified some and we resolved them.

3. Notes are required for the first appearance of the abbreviation, unless it is something well known such as DNA, RNA, etc. Abstract: SN. Please check carefully and use it properly.

Response 3: We checked the manuscript for bbreviations and we explained them at the first appearance in text.

4. What are the advantages of the SNLF in this paper compared to other classic Antioxidants (like tannic acid, 10.1016/j.ijbiomac.2021.12.007)? The comparison and the advantages can be added to the manuscript.

Response 4: We highlighted the advantages of SNLF compared to other classic antioxidants. We also added the suggested reference.

5. Spelling of references must be checked to meet the journal style.

Response 5: We checked the spelling of references and we adapted it according to the journal style.

Reviewer 2 Report

Dear authors,

The manuscript entitled "A Flavonoid-Rich Extract of Sambucus Nigra L. Reduced Lipid Peroxidation in Rat Experimental Model of Gentamicin Ne-phrotoxicity” describes methodologies for to evaluate the capacity of flavonoids rich extract of Sambucus Nigra L. infloresences (SNLF) to inhibit lipoperoxidation (evaluated by malondialdehyde and antioxidant enzymes activity) in GM-induced nephrotoxicity by HPLC, on 4 Wistar male rats groups. It presents scientific relevance for the area of Medicine, Chemistry, and Natural products area.

After consulting www.sciencedirect.com and https://pubmed.ncbi.nlm.nih.gov/, publications were found for some authors involving the theme. The language (English) are satisfactory (I suggest the final revision)! However, you need to change some details/information in the abstract, Introduction, Material and Methods, Results, discussion and “conclusions”.

  1. Abstract: Adequate, but I suggest add information:

- Page 1, line 38: to replace “mg/kg” by “mg Kg-1”and, throughout the manuscript.

- Page 1. Line 44: to replace “GM-SN group” by “GM + SN group”

- In Keywords: “elderflower” and “polyphenols” do not appear in the abstract or title of the manuscript.

- I suggest inserting information about the ranges of concentrations of phenolic acids and flavonoids identified and quantified by HPLC.

- I suggest, at the end of the abstract, highlighting the "innovative" proposal of the method, as well as the advantages / disadvantages.

  1. Introduction section: It is well written, but I suggest:

- Some paragraphs are too short! I suggest joining them, for better understanding and continuity of ideas.

- As the article involves, also, the development and validation of the analytical method, I suggest including information on analytical techniques and more references on the determination of phenolic bioactives and in biological samples (plants in general, or for the reported species) by HPLC. I suggest some references:

- Determination of bioactive phenolics in herbal medicines containing Cynara scolymus, Maytenus ilicifolia Mart ex Reiss and Ptychopetalum uncinatum by HPLC-DAD (Microchemical Journal - volume 135, November 2017, Pages 10-15 – DOI: https://doi.org/10.1016/j.microc.2017.07.009)

- Evaluation of multielement/proximate composition and bioactive phenolics contents of unconventional edible plants from Brazil using multivariate analysis techniques (Food Chemistry, Volume 363, 30 November 2021, 129995 – DOI: https://doi.org/10.1016/j.foodchem.2021.129995)

- Insert “Figure 1” closer to the citation in the text!

- I suggest at the end of the introduction, I suggest highlighting the "innovative" proposal of the method, as well as the advantages / disadvantages.

  1. Materials and Methods section:

- Pages 3-4, in “Quantitative analysis of phenolic compounds in elderflowers” section: I request more details on quantitative analysis, such as: concentration range, analytical validation (precision, accuracy, LOD, LOQ, etc.).

- Page 4, lines 144 and 147, in “Quantitative analysis of phenolic compounds in elderflowers” section: to replace “ml/min” and “mg/mL” by “mL min-1” and “mg mL-1” and, throughout the manuscript.

- Page 4, line 153, in “Protocol for obtaining elderflower extract for in vivo administration” section: to replace “ml” by “mL” and, throughout the manuscript. Was this protocol based on any references? If yes, add!

- Page 4, in “2.4. Experimental design” section: Why was the 10-day period selected? Any specific protocol? I suggest choosing to keep table 1 or the information in the text! It's repeated!

- Page 4, lines 175-176, in “2.4. Experimental design” section: the authors wrote “….(corresponding to 2.3 mg rutin/kg b.w., 18 μg CA /kg b.w. and 7.5 μg FA /kg b.w.) by gavage daily)”…. How were the concentrations of phenolic bioactives in the extract determined? Are the results of quantitative analysis by HPLC? If so, they should be indicated in the "results and discussion" section.

  1. Results section:

Wouldn't it be more interesting to combine the "results” with the "discussion" to better describe the findings and compare them with other works published in the literature? I strongly suggest this union!

- Page 6, in “Table 2”: I suggest reformulation in table 2. The values ​​of "standard deviations" can be after the mean value (X ± sd)... I suggest inserting the detection (LD) and quantification (LQ) limits values ​​of the analytes. The compounds (vanillic and p-coumaric acids and, luteolin) are present in the chromatogram! Not detected/quantified? Therefore, it is important to present the LD and LQ data! The authors wrote in page 6, line 244 (…and other components in low concentrations”)??!!

- Page 6, lines 242-243: to replace “ug/ml” by “µg mL -1” and, throughout the manuscript. Do these data match the information in Page 4, lines 175-176? “….(corresponding to 2.3 mg rutin/kg b.w., 18 μg CA /kg b.w. and 7.5 μg FA /kg b.w.)???

- Page 7-8, in Figures 3,4 and 5: to write “Sambucus Nigra” in italic and, throughout the manuscript!

  1. Discission section:

In this section there is a broad theoretical discussion of the topic. Very good! However, a relationship must be made with the results obtained! I suggest joining the sections "results" and "discussion", and taking this theoretical approach and discussing the obtained data, based on the scientific literature.

- Page 9, lines 297-302: to insert reference for support information!

- I suggest, at the end of the "results and discussion", to write a paragraph summarizing the findings and their impacts on the research proposal.

  1. Conclusion: Adequate, but I suggest to indicate disadvantages/limitations of the method and the study! Perhaps, to highlight the text in the 'Limits of the study' section.
  2. Table and Figures: Adequate! Please, see proposed suggestions!
  3. References: Please, check if the references are in accordance with the journal's rules.

Author Response

Response to Reviewer 2 Comments

  1. Abstract: Adequate, but I suggest add information:

- Page 1, line 38: to replace “mg/kg” by “mg Kg-1”and, throughout the manuscript.

Response: We replaced it throughout the manuscript.

- Page 1. Line 44: to replace “GM-SN group” by “GM + SN group”

Response: We replaced it throughout the manuscript.

- In Keywords: “elderflower” and “polyphenols” do not appear in the abstract or title of the manuscript.

Response: We added ”elderflower” to the abstract and we removed ”polyphenols” from the Keywords.

- I suggest inserting information about the ranges of concentrations (+/_ DEV ST)of phenolic acids and flavonoids identified and quantified by HPLC.

Response: We added the information in the abstract.

- I suggest, at the end of the abstract, highlighting the "innovative" proposal of the method, as well as the advantages / disadvantages.

Response: We highlighted the information suggested at the end of the abstract.

  1. Introduction section: It is well written, but I suggest:

- Some paragraphs are too short! I suggest joining them, for better understanding and continuity of ideas.

Response: We joined some of the paragraphs from the ”Introduction” section.

- As the article involves, also, the development and validation of the analytical method, I suggest including information on analytical techniques and more references on the determination of phenolic bioactives and in biological samples (plants in general, or for the reported species) by HPLC. I suggest some references:

- Determination of bioactive phenolics in herbal medicines containing Cynara scolymus, Maytenus ilicifolia Mart ex Reiss and Ptychopetalum uncinatum by HPLC-DAD (Microchemical Journal - volume 135, November 2017, Pages 10-15 – DOI: https://doi.org/10.1016/j.microc.2017.07.009)

- Evaluation of multielement/proximate composition and bioactive phenolics contents of unconventional edible plants from Brazil using multivariate analysis techniques (Food Chemistry, Volume 363, 30 November 2021, 129995 – DOI: https://doi.org/10.1016/j.foodchem.2021.129995)

Response: We have included supplimentary information and references on analytical techniques, but we moved them to the ”Materials and methods” section.

- Insert “Figure 1” closer to the citation in the text!

Response: We have placed ”Figure 1” closer to the citation in the text.

- I suggest at the end of the introduction, I suggest highlighting the "innovative" proposal of the method, as well as the advantages / disadvantages.

Response: We highlighted the information suggested at the end of the introduction.

  1. Materials and Methods section:

- Pages 3-4, in “Quantitative analysis of phenolic compounds in elderflowers” section: I request more details on quantitative analysis, such as: concentration range, analytical validation (precision, accuracy, LOD, LOQ, etc.).

Response: We added suplimentary details on quantitative analysis.

- Page 4, lines 144 and 147, in “Quantitative analysis of phenolic compounds in elderflowers” section: to replace “ml/min” and “mg/mL” by “mL min-1” and “mg mL-1” and, throughout the manuscript.

Response: We replaced them throughout the manuscript.

- Page 4, line 153, in “Protocol for obtaining elderflower extract for in vivo administration” section: to replace “ml” by “mL” and, throughout the manuscript. Was this protocol based on any references? If yes, add!

Response: We added the references.

- Page 4, in “2.4. Experimental design” section: Why was the 10-day period selected? Any specific protocol? I suggest choosing to keep table 1 or the information in the text! It's repeated!

Response: GM was administered daily for a period of ten days, in accordance with previous experimental models, and with the recommendations made by the medical guidelines. We chose to remove Table 1 and to keep the information in the text.

- Page 4, lines 175-176, in “2.4. Experimental design” section: the authors wrote “….(corresponding to 2.3 mg rutin/kg b.w., 18 μg CA /kg b.w. and 7.5 μg FA /kg b.w.) by gavage daily)”…. How were the concentrations of phenolic bioactives in the extract determined? Are the results of quantitative analysis by HPLC? If so, they should be indicated in the "results and discussion" section.

Response: Due to an error of expression, the text did not highlight the difference between the HPLC analyzed elderflower extract and the elderflower extract for in vivo administration. As the amount of elderflower in 1 mL SN extract administered to the rats was different from the amount of elderflower in 1 mL SN extract analysed by HPLC, due to different preparation protocols, the amount of active compounds was calculated based on the amount of elderflower in 1 mL SNLF extract administered to the rats. We have inserted the method of calculating the main antioxidant compounds in the ”Materials and methods”, as well as in ”Results” section. The amount of antioxidant compounds in the elderflower extract administered in vivo was calculated per 1 gram of elderflower, according to the laws of proportionality (the Rule of Three).

  1. Results section:

Wouldn't it be more interesting to combine the "results” with the "discussion" to better describe the findings and compare them with other works published in the literature? I strongly suggest this union!

Response: We combined the ”Results” and ”Discussion” sections.

- Page 6, in “Table 2”: I suggest reformulation in table 2. The values ​​of "standard deviations" can be after the mean value (X ± sd)... I suggest inserting the detection (LD) and quantification (LQ) limits values ​​of the analytes. The compounds (vanillic and p-coumaric acids and, luteolin) are present in the chromatogram! Not detected/quantified? Therefore, it is important to present the LD and LQ data! The authors wrote in page 6, line 244 (…and other components in low concentrations”)??!!

Response: Table 2 became Table 1 and was reformulated by combining the mean value with the standard deviation. We preferred to insert the detection and quantification limits values ​​of the analytes in the main text, at the section ”Materials and methods”. Representative polyphenolic compounds for elderflowers were analyzed by HPLC and there were standards for them. Quercetin 3-rutinoside (rutin) was the representative flavonol in the elderflower blooming stage.

- Page 6, lines 242-243: to replace “ug/ml” by “µg mL -1” and, throughout the manuscript. Do these data match the information in Page 4, lines 175-176? “….(corresponding to 2.3 mg rutin/kg b.w., 18 μg CA /kg b.w. and 7.5 μg FA /kg b.w.)???

Response: We replaced ”ug/ml” by “µg mL-1” throughout the manuscript. By error, in the experimental model the administered amount of different elderflower compounds was reported to the SNLF extract analysed by HPLC. As the amount of elderflower in 1 mL SNLF extract administered to the rats was different from the amount of elderflower in 1 mL SNLF extract analysed by HPLC, due to different preparation protocols, the amount of active compounds was calculated based on the amount of elderflower in 1 mL SNLF extract administered to the rats. We corrected the values in the manuscript and we presented the way they have been calculated.

- Page 7-8, in Figures 3,4 and 5: to write “Sambucus Nigra” in italic and, throughout the manuscript!

Response: We wrote ”Sambucus Nigra” in italics throughout the manuscript.

  1. Discussion section:

In this section there is a broad theoretical discussion of the topic. Very good! However, a relationship must be made with the results obtained! I suggest joining the sections "results" and "discussion", and taking this theoretical approach and discussing the obtained data, based on the scientific literature.

Response: We combined the ”Results” and ”Discussion” sections.

- Page 9, lines 297-302: to insert reference for support information!:

Pizzino G, Irrera N, Cucinotta M, et al. Oxidative Stress: Harms and Benefits for Human Health. Oxid Med Cell Longev. 2017;2017:8416763. doi:10.1155/2017/8416763

Response: We inserted the suggested reference.

- I suggest, at the end of the "results and discussion", to write a paragraph summarizing the findings and their impacts on the research proposal.

Response: We wrote a summarizing paragraph at the end of the „Results and Discussion” section.

  1. Conclusion: Adequate, but I suggest to indicate disadvantages/limitations of the method and the study! Perhaps, to highlight the text in the 'Limits of the study' section.

Response: The limits of the study are presented in a special sub-section in the end of the ”Discussion” section. They have been supplimented.

  1. Table and Figures: Adequate! Please, see proposed suggestions!

Response: We performed the suggested adaptations.

  1. References: Please, check if the references are in accordance with the journal's rules.

Response: We checked the references and we adapted them according to the journal’s rules.

Reviewer 3 Report

The work is directed to the search of antioxidants which can reduce the nefrotoxicity of gentamicin, a widely-used antibiotic. For this purpose the authors offer elderflower extract, basing on their previous results, which had shown that application of elderflower extract along with gentamicin reduced the tissue damage. Present work was designed very similarly to the previous one, but it was aimed to give further insight into the mechanism of action and to prove the antioxidant properties of elderflower extract; this suggestion was proved successfully via in vivo experiments. The work is actual, consistent and, generally, well-written, yet, a number of questions should be addressed and the paper may be published only after major revision.

The following remarks should be considered prior to the publication.

  • Full list of abbreviation should be given in the beginning of the article to make the text readable.
  • Figure 1. It would be better to remove the picture of elderberry and instead enlarge the chemical structures of flavonols. The numbers of compounds should be given in parentheses: rutin (1).
  • Figures 3 and 6. The description of the effects, which duplicate the main text, should be removed from the captions.
  • Was the elderflower harvested in 2017 or in 2018? Two dates are given in the text.
  • Toxicity of individual compounds and elderflower extract in the concentrations, used in the study, should be given.
  • A number of known antioxidants have been investigated as protective agents when using gentamicin. The comparison of the effect of elderflower extract with previously described protective agents should be given.
  • It is known that the use of antioxidants together with gentamicin may result in reducing of antibacterial effect of the antibiotic. This point should be commented.
  • A number of peaks on chromatogram of SNLF sample are not assigned. What compounds do they correspond? May they also act as antioxidants? Is it correct to describe rutin as the main active component in the presence of other compounds, some of them, unknown?
  • Are there any data of the action of individual phenol compounds in similar conditions? Is there any synergism in their action?
  • The authors state: “Thus, it is particularly important to determine the composition of plant extracts used in each experimental model and to correlate the therapeutic effects with the main compounds found in the studied sample.” From this point of view, does the composition vary significantly? May if totally change the results of experiment?
  • Is the effect of antioxidants time-dependent?
  • May the quantitative values of effect (as EC50) be given?
  • The mechanism of antioxidant action of flavonoids should be given on the example of rutin.
  • The discussion should be considerably shortened. Extensive description of the oxidative stress, gentamicin action, action of antioxidant enzymes would look better in a textbook. In the article it would be better to focus on the results obtained and their comparison with previously described ones.

Author Response

Response to Reviewer 3 Comments

  • Full list of abbreviation should be given in the beginning of the article to make the text readable.

Response: We added the abbreviation list at the end of the manuscript, before „References”.

  • Figure 1. It would be better to remove the picture of elderberry and instead enlarge the chemical structures of flavonols. The numbers of compounds should be given in parentheses: rutin (1).

Response: We removed the picture of elderberry and instead enlarged the chemical structures of flavonols. The numbers of compounds have been given in parentheses.

  • Figures 3 and 6. The description of the effects, which duplicate the main text, should be removed from the captions.

Response: We removed from the Figures the description of the effects, which were duplicating the main text.

  • Was the elderflower harvested in 2017 or in 2018? Two dates are given in the text.

Response: We corrected the error concerning elderflower harvesting year. The correct year is 2018.

  • Toxicity of individual compounds and elderflower extract in the concentrations, used in the study, should be given.

Response: Information about toxicity of the individual compounds of the elderflower extract was added in the manuscript.

  • A number of known antioxidants have been investigated as protective agents when using gentamicin. The comparison of the effect of elderflower extract with previously described protective agents should be given.

Response: Comparison between the effect of elderflower extract and other previously described protective agents was added.

  • It is known that the use of antioxidants together with gentamicin may result in reducing of antibacterial effect of the antibiotic. This point should be commented.

Response: The point was commented in the „Results and Discussion” section, also identified as a new research direction.

  • A number of peaks on chromatogram of SNLF sample are not assigned. What compounds do they correspond? May they also act as antioxidants? Is it correct to describe rutin as the main active component in the presence of other compounds, some of them, unknown?

Response: Representative polyphenolic compounds for elderflowers were analyzed by HPLC and there were standards for them. Quercetin 3-rutinoside (rutin) was the representative flavonol in the elderflower blooming stage.

  • Are there any data of the action of individual phenol compounds in similar conditions? Is there any synergism in their action?

Response: Data were added and discussed in the „Results and Discussion”.

  • The authors state: “Thus, it is particularly important to determine the composition of plant extracts used in each experimental model and to correlate the therapeutic effects with the main compounds found in the studied sample.” From this point of view, does the composition vary significantly? May if totally change the results of experiment?

Response: We added as another limit of study the multifactorial environmentally induced variability of elderflowers composition, which means that the results of the present study can only be extrapolated to similar extracts.

  • Is the effect of antioxidants time-dependent?

Response: We noted in the article that, in order to improve efficiency, curves for time-dependent activity of the whole SN extract should be also calculated in the future.

  • May the quantitative values of effect (as EC50) be given?

Response: No. The aim of the study was not to establish the dose-effect relationship, but to verify the involvement of the antioxidant mechanism in the nephroprotective effect of the elderflower extract. Of course, many other research directions are open and we are grateful that many of them have been highlighted to us by your review. In accordance with your observation, we noted in the article that the value of EC50 should be calculated in the next studies.

  • The mechanism of antioxidant action of flavonoids should be given on the example of rutin.

Response: There is a sub-chapter in „Results and Discussion” focussing this point, where rutin’s role is highlighted.

  • The discussion should be considerably shortened. Extensive description of the oxidative stress, gentamicin action, action of antioxidant enzymes would look better in a textbook. In the article it would be better to focus on the results obtained and their comparison with previously described ones.

Response: Discussion has been shortened.
